# Threat Assessment Method of Low Altitude Slow Small (LSS) Targets Based on Information Entropy and AHP

**DOI:** 10.3390/e23101292

**Published:** 2021-09-30

**Authors:** Ruining Luo, Shucai Huang, Yan Zhao, Yafei Song

**Affiliations:** Air and Missile Defense College, Air Force Engineering University, Xi’an 710051, China; hsc67118@126.com (S.H.); zytyler@163.com (Y.Z.); yafei_song@163.com (Y.S.)

**Keywords:** LSS targets threat assessment, analytic hierarchy process, information entropy

## Abstract

In order to deal with the new threat of low altitude slow small (LSS) targets in air defense operations and provide support for LSS target interception decision, we propose a simple and reliable LSS target threat assessment method. Based on the detection capability of LSS targets and their threat characteristics, this paper proposes a threat evaluation factor and threat degree quantization function in line with the characteristics of LSS targets. LSS targets not only have the same threat characteristics as traditional air targets but also have the unique characteristics of flexible mobility and dynamic mission planning. Therefore, we use analytic hierarchy process (AHP) and information entropy to determine the subjective and objective threat factor weights of LSS targets and use the optimization model to combine them to obtain more reliable evaluation weights. Finally, the effectiveness and credibility of the proposed method are verified by experimental simulation.

## 1. Introduction

Threat assessment is an estimate of the lethality of enemy forces and the degree of threat to us [1]. The construction of target threat assessment model usually includes two key contents: one is to reasonably select target threat assessment factors and form quantitative evaluation indicators; the other is to determine the weight of each assessment factor.

In the field of air defense, LSS targets have the characteristics of low altitude flight and slow speed, and they are not easy to be found, which brings great threats and challenges to key protection targets. It is of great significance to judge the incoming targets in time for low altitude air defense. Taking the typical LSS target of an unmanned aerial vehicle (UAV) swarm as an example, it can be seen from the UAV combat concept, development planning and research projects published in recent years [2,3,4,5,6], that a UAV swarm is composed of dozens to hundreds of UAVs with low cost, small size and limited functions equipped with a variety of mission loads. The single UAV platform constituting the UAV swarm can be isomorphic or heterogeneous UAVs, and the types and number of UAVs in the swarm are also different. Therefore, the future battlefield will be faced with large-scale drone swarms with different types and threats. In the anti-swarm operation, it is important be able to accurately evaluate the threat value of each UAV. While studying the methods and means of anti-UAV, we should also pay attention to how to identify and evaluate the threat degree of UAVs of different types and states on the basis of existing detection means, so as to provide auxiliary decision support for anti-UAV operations and UAV cluster confrontation.

At present, there are many methods that can be used to determine the weight of evaluation factors. According to the data sources used in determining the weight, it can be divided into subjective weighting method, objective weighting method and combined weighting method.

Subjective weighting methods are an older and more mature weighting method, such as Delphi Method, Best-Worst Method, AHP, Analytic Network Process, etc., which have been widely used in many fields. In the latest research results, the authors of [7] improved the Best-Worst Method and proposed a linear Best-Worst Method and Euclidean Best-Worst Method. Reference [8] is based on AHP and adopts consistent fuzzy preference relations to construct the decision matrix. Both documents improve the quality of expert decision-making in multi-criteria decision-making. Reference [9] introduced the D number theory into ANP to better adapt to the uncertainty of expert judgment. Reference [10] proposed a Bayesian cosine maximization method to modify the comparison matrix to obtain more accurate estimates of the priority vectors. Among them, AHP is still the most concise and practical decision-making method, although there are problems in rank reversal, comparison scale, priorities derivation method, and so on [11,12]. However, when there are fewer decision variables, such problems are not prominent, and eigenvalues and eigenvectors can still be obtained accurately. In addition, many new improved methods are also based on AHP. Therefore, in this new application direction of LSS target threat assessment, this article still applies AHP to explore the applicability of AHP and improved methods based on this in the field of LSS target threat assessment.

Different from the traditional air target, the LSS target represented by UAV swarm has the problems of difficult detection and incomplete information. In addition, the UAV swarm has flexible maneuverability and dynamic mission planning capabilities. It is difficult to make accurate judgments on its combat capabilities and attempts using traditional experience and knowledge. The subjective weighting methods commonly used in target threat estimation need large-scale auxiliary systems and a priori knowledge base. At present, the research on UAV swarm warfare is still in the preliminary stage, and it is difficult to obtain accurate evaluation results based on such methods. Therefore, we hope to find an effective objective weighting method to assist in order to obtain more reliable evaluation results. Among them, the entropy method determines the weight according to the index variability, and the weight is adjusted in real-time, which can sensitively find the “dissimilarity” in the cluster, highlight the high threat targets that need to be paid attention to in the cluster, and can be suitable for the threat assessment of LSS targets.

In this paper, the detection capability of LSS targets is analyzed and the threat characteristics of LSS targets are discussed. Based on these, the threat assessment factors and quantification methods suitable for LSS targets are proposed. AHP and entropy method are introduced into LSS target threat assessment, and an optimization model is established to optimize and fuse the weights determined by them to obtain a more reliable assessment weight. The method proposed in this paper can simply and reliably complete the LSS target threat assessment, provide a reference for the interception decision of LSS target and provide a new reference for the research of LSS target threat assessment.

The structure of this paper is as follows. In Section 2, we select the LSS target threat assessment factors and the quantitative function of each assessment factor. In Section 3, an LSS target threat assessment model is proposed. In Section 4, the LSS target threat assessment process is introduced. In the fifth section, we simulate and verify the method in this paper. In Section 6, we conclude the paper and show future directions.

## 2. Selection and Quantification of Threat Assessment Factors for LSS Targets

The selection of LSS target threat assessment factors needs to fully consider the current detection technology and capability and the ability of different characteristic factors to reflect the degree of target threat.

Through extensive access to data, the detection capabilities of the current two mainstream detection methods of radar and infrared to LSS targets [13,14,15,16,17,18] are obtained, as shown in Table 1.

It can be seen that both radar and infrared can effectively detect LSS targets. Although it is difficult for a single technical means to achieve stable tracking of weak and small targets in complex battlefield environments, it can be considered that it is capable of all-weather autonomous detection of LSS targets within a certain distance under the condition of multi-sensor fusion based on the two methods supplemented by radio spectrum detection and laser detection.

When the target space status information can be obtained, the traditional threat factors such as target speed, entry angle, height and distance from our defense position can be selected. These factors can well reflect the target mobility, interception urgency and other information, which is also applicable to LSS targets.

Heterogeneity is a typical feature of UAV swarm. Different UAVs in the swarm perform different tasks. Large UAVs usually have a greater load, stronger attack, detection, communication, decision-making or interference capabilities, and pose a greater threat to our targets. Therefore, the imaging area of the target on the radar can better reflect the threat degree of the target.

LSS target recognition is also the focus of target detection. Target recognition mainly collects UAV images through imaging technology for comparative analysis and feature extraction to realize target recognition. A target detection algorithm based on deep learning is the focus of current research. Many scholars have used a neural network algorithm to effectively identify the type of LSS targets platform [15,19], and the maximum recognition distance based on visible light imaging can reach about 0.8 km [13].

To sum up, the angle, speed, height and distance threat factors that can reflect the target’s dynamic threat capability are selected as the threat evaluation indicators of LSS targets together with RCS and target type threat factors that can reflect the target’s static threat capability. The above six impact factors have different impacts on the target threat degree, so it is necessary to quantify the indicators system as follows.

Speed threat factor

The faster the target speed, the stronger the maneuverability, the shorter the time to fly to our defense position, and the greater the threat to us. Therefore, we propose the threat function of the velocity factor, as shown in Figure 1.

The corresponding mathematical expression is:(1)$$S_{v}=\left\{ \begin{array}{ll} 0.2 & v<0.2Ma\\ \frac{{\left(v-0.2\right)}^{2}}{0.5\times (0.6-0.2)}+0.2 & \;0.2Ma\leq v<0.6Ma\\ 1 & v>0.6Ma \end{array}\right.$$

2.Angle threat factor

The target angle threat can be described by the target entry angle. The target entry angle refers to the angle between the connecting line between the target and our array and the target flight direction. The smaller the entry angle is, the more obvious the target attack intention is and the greater the threat to us. Therefore, we propose the threat function of the angle factor, as shown in Figure 2.

The corresponding mathematical expression is:(2)$$S_{\alpha }=\left\{ \begin{array}{ll} 1-{\left(\frac{\alpha }{90}\right)}^{2} & -90\leq \alpha \leq 90\\ 0 & \alpha >90\cup \alpha <-90 \end{array}\right.$$

3.Height threat factor

The lower the target height, the easier it is to avoid radar detection, which is not conducive to the shooting of air defense weapons. At the same time, the target with a lower height can act on our position faster. On the contrary, the target in too high airspace is difficult to directly perform the established mission, that is, the target threat degree increases with the decrease in target height. Therefore, we propose the threat function of the height factor, as shown in Figure 3.

The corresponding mathematical expression is:(3)$$S_{h}=\left\{ \begin{array}{ll} 1 & h<100\\ e^{\frac{100-h}{500}} & h\geq 100 \end{array}\right.$$

4.Distance threat factor

The closer the target is to our array, the easier it is to carry out direct action, and the shorter the response time left to our defense system. Therefore, we propose the threat function of the distance factor, as shown in Figure 4.

The corresponding mathematical expression is:(4)$$S_{r}=\left\{ \begin{array}{ll} 1 & h<100\\ e^{\frac{2-r}{20}} & h\geq 100 \end{array}\right.$$

5.RCS threat factor

The target RCS value fluctuates with the change of motion state, but it is usually proportional to the target size. The larger the RCS, the stronger the load and maneuverability, and the greater the threat to our array. Therefore, we propose the threat function of the RCS factor, as shown in Figure 5.

The corresponding mathematical expression is:(5)$$S_{R}=\left\{ \begin{array}{ll} 0.2 & RCS<0.01\\ 0.2+\frac{{\left(RCS-0.01\right)}^{2}}{1.25\times 0.09^{2}} & 0.01\leq RCS<0.1\\ 1 & RCS\geq 0.1 \end{array}\right.$$

6.Type threat factor

At present, the types of UAV platforms for UAV swarm operations are mainly divided into rotor platforms and fixed-wing platforms. The type of target platform is closely related to its operational tasks. For example, the rotor UAV flies slowly and mainly performs reconnaissance and jamming tasks. The fixed-wing UAV flies faster and can perform more critical reconnaissance, communication and fire attack tasks. Therefore, it is considered that the threat degree of the fixed-wing UAV is higher than that of the rotor UAV. The quantitative method of threat degree according to the type of target platform is:(6)$$S_{c}=\left\{ \begin{array}{ll} 0.5 & \mathrm{Rotor}\;\mathrm{UAV}\\ 1 & \mathrm{Fixed}\;\mathrm{wing}\;\mathrm{UAV} \end{array}\right.$$

## 3. LSS Target Threat Assessment Model

### 3.1. Determination of Subjective Weight by AHP 

An analytic hierarchy process is a multi-objective decision-making analysis method combining qualitative and quantitative analysis. It uses expert experience to compare the importance of the participating factors, constructs a comparison matrix, determines the relative importance of each factor in the hierarchy, and then constructs a judgment matrix to quantitatively determine the weight of each threat factor.

The determination steps of each index weight based on AHP are as follows [1,14,20,21,22].

**Step** **1**: According to the selected threat factors, establish the analytic hierarchy process model diagram, as shown in Figure 6.

**Step** **2**: Build a comparison matrix. The 9-level proportional scaling method is used to compare the factors in the same layer and construct the judgment matrix $A={\left(a_{ij}\right)}_{6\times 6}$, The basic evaluation criteria include “equally important”, “slightly important”, “quite important”, “extremely important” and “absolutely important”, which correspond to 1, 3, 5, 7 and 9 in the 9-level scale respectively, and the compromise values of each scale are represented by 2, 4, 6 and 8.

In order to obtain expert knowledge, draw an expert scoring table for the pairwise comparison of the six indicators of target speed, target entry angle, target height, target distance, target RCS and target type. Three teachers of the subject of the college are invited to score, using three scoring tables; in addition, extract the expert knowledge from the literature [20,21] to produce two scoring tables. Synthesize five expert quantitative score tables, average and round them to produce the judgment matrix as shown in Table 2.

**Step** **3**: Calculate indicators of weight value and consistency test. The weight value of each indicator can be obtained by normalizing the eigenvector corresponding to the maximum eigenvalue of judgment matrix A. In practical engineering applications, the comparison matrix is required to approximately meet the basic consistency, and the judgment matrix is obtained according to people’s subjective judgment, which inevitably has estimation error. Therefore, the consistency test of ranking should be carried out.

Using the consistency indicators $C.I.=(\lambda_{\max}-n)/(n-1)$, and the consistency ratio C.R.=C.I./R.I. for consistency test, when the number of indicators is six, $C.I.=(\lambda_{\max}-n)/(n-1)$ is generally selected. When C.R.=0, it shows that the comparison matrix strictly satisfies the basic consistency; When C.R.<0.1, it can be considered that the comparison matrix approximately satisfies the basic consistency.

The maximum eigenvalue of the comparison matrix is $\lambda_{\max}=6.0271$ and the consistency ratio is C.R.=0.004<0.1, so the comparison matrix approximately meets the basic consistency. The eigenvector corresponding to the maximum eigenvalue $\lambda_{\max}$ of the comparison matrix is [0.4022  0.0813  0.2078  0.7639  0.4022  0.2078], which is normalized to obtain the weight value of each indicator as $W_{A}={\left[0.194\;\;0.039\;\;0.101\;\;0.370\;\;0.195\;\;0.101\right]}^{T}$.

### 3.2. Determination of Objective Weight by Entropy Method 

After determining the subjective weight based on expert experience, the information entropy theory is introduced into the determination of target attribute weight of air target threat assessment, which can make the determination of weight more objective and reflect the impact of each attribute on target threat assessment.

In order to introduce the concept of information entropy and establish the threat assessment model of cluster targets, the definition of information entropy is given as follows:

**Definition** **1**([23]). *The mathematical expectation of self-information is defined as the average self-information of the source. If an object may have* n
*states and the probability of each state is* $\left(p_{1},p_{2},\cdot \cdot \cdot ,p_{n}\right)$*, the information entropy of the object is defined as*:


(7)
$$H(X)=E\left[\log\frac{1}{P(a_{i})}\right]=-\sum_{i=1}^{q}P(a_{i})\log P(a_{i})$$


The determination steps of each indicator weight based on information entropy are as follows [23,24,25,26,27,28,29,30].

**Step** **1**: Construct decision matrix. Assuming that there are m air raid targets, n threat factors are selected as the evaluation indicator, and the decision matrix $A={\left(a_{ij}\right)}_{m\times n}$ is constructed according to the threat degree quantification method of each factor, where *a_ij_* is the quantitative value of the *i*-th target under the *j*-th threat factor.

Normalize the decision matrix A to obtain $\tilde{A}={\left({\tilde{a}}_{ij}\right)}_{m\times n}$, where
(8)$${\tilde{a}}_{ij}=a_{ij}/\sum_{i=1}^{m}a_{ij}$$

**Step** **2**: Calculate the information entropy of each indicator. Information entropy is a measure of the disorder degree of the system. The smaller the information entropy of an attribute indicator, the greater the variation degree of the indicator value, the greater the amount of information, and the greater the weight in the evaluation [25]. Therefore, the information entropy $H_{j}$ of the j-th evaluation indicator to the target is defined as:


(9)
$$H_{j}=-\frac{1}{\ln m}\sum_{i=1}^{m}{\tilde{a}}_{ij}\ln{\tilde{a}}_{ij}$$


**Step** **3**: Calculate the weight of each indicator. According to the information entropy $H_{j}$ of each indicator, the weight value $W_{E}={\left[w_{1},w_{2},\cdot \cdot \cdot ,w_{m}\right]}^{T}$ of each indicator is obtained, where


(10)
$$w_{j}=\frac{1-H_{j}}{\sum_{k=1}^{m}\left(1-H_{k}\right)}$$


### 3.3. Determination of Final Indicators Weight and Calculation of Target Threat Value

The AHP based on expert experience and the entropy method based on objective information reflect the weight of target attributes in threat estimation from different angles. Therefore, based on the above two weight schemes, it is expected to find an optimized weight $W={\left[w_{1},w_{2},\cdot \cdot \cdot ,w_{m}\right]}^{T}$ to take values between subjective and objective weights in an optimal way, so as to obtain maximum reliability.

The core idea of the optimization model is: in order to fully extract the reasonable part of the subjective and objective weights and realize the organic unity of the objective characteristics of expert experience and battlefield data, the deviation between the combined weight vector and the evaluation value vector corresponding to each weight vector should be as small as possible. Thus, the following optimization model is constructed [31]:(11)$$\begin{array}{c} \min\;z=\sum_{j=1}^{n}\left[\alpha {\left(w_{j}-w_{Aj}\right)}^{2}+(1-\alpha ){\left(w_{j}-w_{Ej}\right)}^{2}\right]\\ s.t.\;\;\;\sum_{j=1}^{n}w_{j}=1,\;w_{j}\geq 0\;\;\;(j=1,2,\cdot \cdot \cdot ,n) \end{array}$$
where: $w_{Aj}$ and $w_{Ej}$ are the subjective and objective weights of the j-th indicator obtained by analytic hierarchy process and entropy method, respectively, and $w_{j}$ is the final weight of the j-th indicator obtained by optimization. α is the partiality coefficient. When the expert experience is more accurate and reliable, α takes the larger value, on the contrary, α takes the smaller value.

According to the optimized weight obtained, the threat degree F of LSS target can be obtained by multiplying the normalized target threat degree matrix $\tilde{A}$ and the weight vector W of each evaluation indicator, as:(12)$$F=\tilde{A}W=\left[\begin{array}{cccc} {\tilde{a}}_{11} & {\tilde{a}}_{12} & \cdot \cdot \cdot & {\tilde{a}}_{1n}\\ {\tilde{a}}_{21} & {\tilde{a}}_{22} & \cdot \cdot \cdot & {\tilde{a}}_{2n}\\ \cdot \cdot \cdot & \cdot \cdot \cdot & \cdot \cdot \cdot & \cdot \cdot \cdot \\ {\tilde{a}}_{m1} & {\tilde{a}}_{m2} & \cdot \cdot \cdot & {\tilde{a}}_{mn} \end{array}\right]\left[\begin{array}{c} w_{1}\\ w_{2}\\ \cdot \cdot \cdot \\ w_{n} \end{array}\right]={\left(f_{1},f_{2},\cdot \cdot \cdot ,f_{n}\right)}^{T}$$

## 4. Threat Assessment Process

The information entropy method is combined with AHP to build a method model for threat assessment of LSS targets, as shown in Figure 7.

The LSS target threat assessment process based on AHP and information entropy is mainly divided into two modules. The first is the air information acquisition module. Relying on the target detection and tracking system based on radar and infrared detection means, the four dynamic attributes of target speed, entry angle, height and distance are obtained in real-time. Relying on the detection and identification system based on radar and photoelectric detection means, the two static attribute information of target RCS and platform type are obtained. Then, according to the threat degree calculation function of each attribute, the quantitative threat degree value of each attribute is obtained, and it is normalized to obtain the decision matrix. The second is the weight determination module of each threat evaluation indicator. Based on expert experience and knowledge, the importance of the six selected threat assessment factors is compared and analyzed by analytic hierarchy process, and the corresponding subjective weight vector is obtained. Based on the real-time target feature information, the objective weight vector of each evaluation indicator is calculated by using the method of information entropy. The final weight vector with more credibility is obtained by optimizing the combination of the two. Finally, the decision matrix is multiplied by the corresponding weight vector to obtain the threat assessment results of each target.

## 5. Experiment

Scenario of battlefield environment: assuming that there are five UAVs near our defense position, and all of them are within the action range of our comprehensive detection system, the target characteristic information can be obtained at a certain time, as shown in Table 3.

It is required to conduct threat assessment on the target according to the obtained target characteristic information, so as to provide reference for subsequent interception operations.

Calculate the six threat factors of the target’s speed, entry angle, height, distance, RCS and type according to formulas (1)~(6), and obtain the threat degree matrix of the target:$$A=\left[\begin{array}{cccccc} 0.200 & 0.849 & 0.819 & 0.317 & 0.684 & 0.500\\ 0.250 & 0.627 & 0.497 & 0.670 & 0.289 & 0.500\\ 0.313 & 0.988 & 0.333 & 0.407 & 0.556 & 1\\ 0.513 & 0.927 & 0.670 & 0.861 & 0.358 & 1\\ 0.213 & 0.951 & 0.905 & 0.192 & 0.210 & 0.500 \end{array}\right]$$

Normalize matrix A to obtain matrix $\tilde{A}$:$$\tilde{A}=\left[\begin{array}{cccccc} 0.134 & 0.193 & 0.254 & 0.130 & 0.326 & 0.143\\ 0.168 & 0.143 & 0.154 & 0.274 & 0.138 & 0.143\\ 0.210 & 0.225 & 0.103 & 0.166 & 0.265 & 0.286\\ 0.345 & 0.222 & 0.208 & 0.352 & 0.171 & 0.286\\ 0.143 & 0.217 & 0.281 & 0.078 & 0.100 & 0.143 \end{array}\right]$$

Firstly, the objective weight is calculated based on the method of information entropy, and the information entropy of evaluation indexes is calculated according to formula (9), and the following is obtained:H=[0.958  0.992  0.955  0.923  0.946  0.963]

Then, use formula (10) to calculate the objective weight, and the following is obtained:$$W_{E}={\left[0.165\;\;0.031\;\;0.137\;\;0.307\;\;0.214\;\;0.147\right]}^{T}$$

The subjective weight obtained based on AHP has been given in Section 3.1. The weight fusion is carried out by using the optimization model given in Section 3.3. In order to verify the influence of the value of partiality coefficient α on the results, this paper takes α=0.3 and α=0.7, respectively for simulation verification.

When α=0.3, the obtained results are shown in Figure 8.

The final weight obtained after optimization is
$$W={\left[0.174\;\;0.033\;\;0.126\;\;0.326\;\;0.208\;\;0.133\right]}^{T}$$

According to formula (12), the quantitative value of threat degree of each target is:$$F={\left[0.191\;\;0.190\;\;0.204\;\;0.282\;\;0.133\right]}^{T}$$

The order of target threats is:target 4>target 3>target 1>target 2>target 5
when α=0.7, the obtained results are shown in Figure 9.

The final weight obtained after optimization is
$$W={\left[0.185\;\;0.037\;\;0.119\;\;0.351\;\;0.201\;\;0.115\right]}^{T}$$

The quantitative value of threat degree of each target is:$$F={\left[0.190\;\;0.195\;\;0.204\;\;0.287\;\;0.132\right]}^{T}$$

The order of target threats is:target 4>target 3>target 2>target 1>target 5

It can be seen that the adjustment of partiality coefficient α changes the threat estimation value of each target and the ranking results of target 1 and target 2. The larger α is, the closer the result is to AHP, and the smaller α is, the closer the result is to the result obtained by information entropy method.

From the simulation results, it can be seen that the results obtained by AHP and entropy method are essentially the same, which shows that the two methods have certain rationality and credibility in LSS target threat assessment. The weight obtained after the optimized combination is more neutral, which can absorb the reasonable part of the two methods at the same time, and avoid large deviation caused by the irrationality of one of them. The selection of partiality coefficient α can directly affect the final weight value, and its value can be determined according to the completeness and reliability of the established expert system. To sum up, the analysis can prove that the model has credibility and effectiveness in LSS target threat assessment and has the advantages of being a simple method requiring a small amount of calculation.

## 6. Conclusions and Future Work

Based on expert experience, AHP can make a reasonable evaluation of conventional target attributes and tactical actions. The information entropy method can effectively reflect the variability of target attributes and can quickly respond to target dynamic task planning and maneuver, which is especially suitable for threat estimation when target information is not comprehensive. The model established in this paper comprehensively considers the subjective and objective weights given by the two and fuses them in an optimal way to obtain a more reliable evaluation result. Finally, the credibility and effectiveness of the model are verified by simulation analysis. The target threat degree obtained by this method can provide reference for LSS target interception decision.

Nevertheless, the method proposed in this paper has its limitations: when the evaluation criteria increase, the result of AHP may become worse; when the established expert knowledge base is incomplete, the reliability of the evaluation will decline.

As for future work, with the deepening of LSS target operation research and the enrichment of expert knowledge base, we can further select new threat assessment factors, further expand the structure of threat assessment factor quantization function and optimize its parameters. We can further improve AHP to better adapt to the increase in evaluation factors.

## Figures and Tables

**Figure 1 entropy-23-01292-f001:**
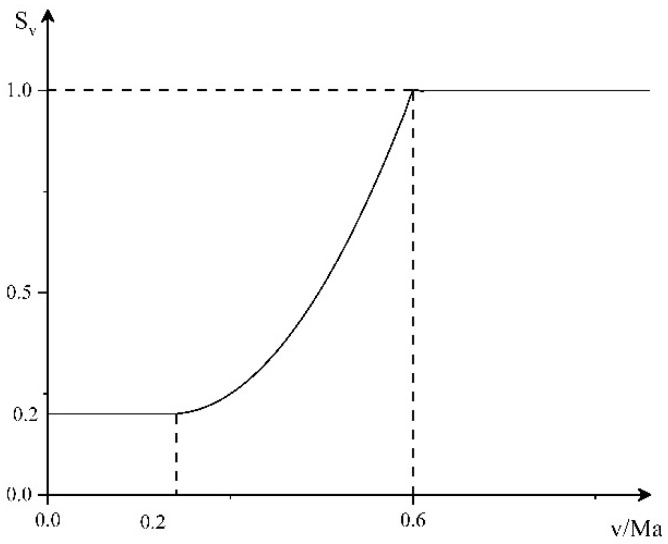
Speed threat function.

**Figure 2 entropy-23-01292-f002:**
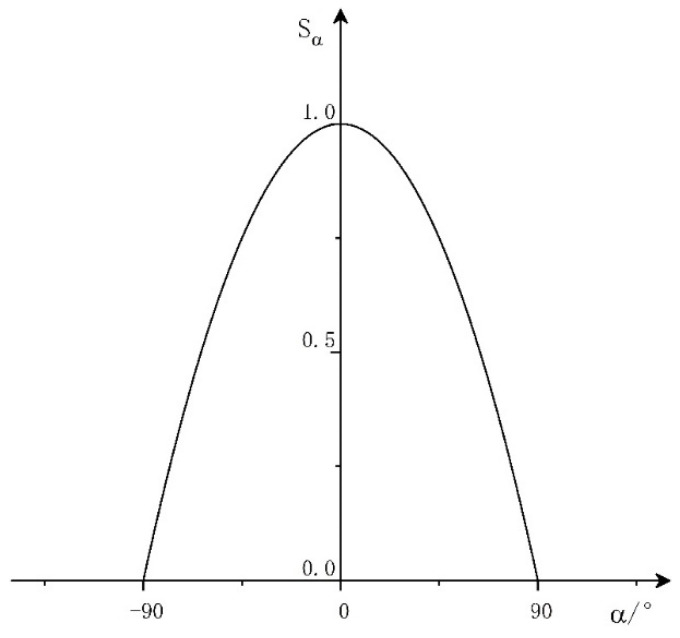
Angle threat function.

**Figure 3 entropy-23-01292-f003:**
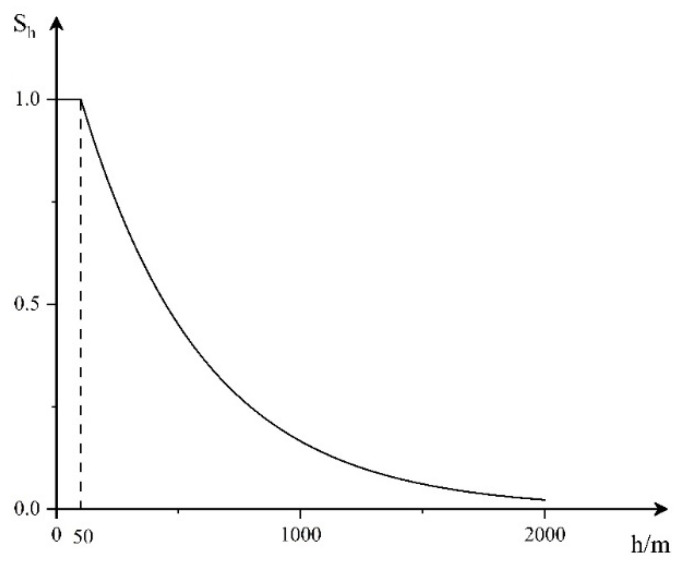
Height threat function.

**Figure 4 entropy-23-01292-f004:**
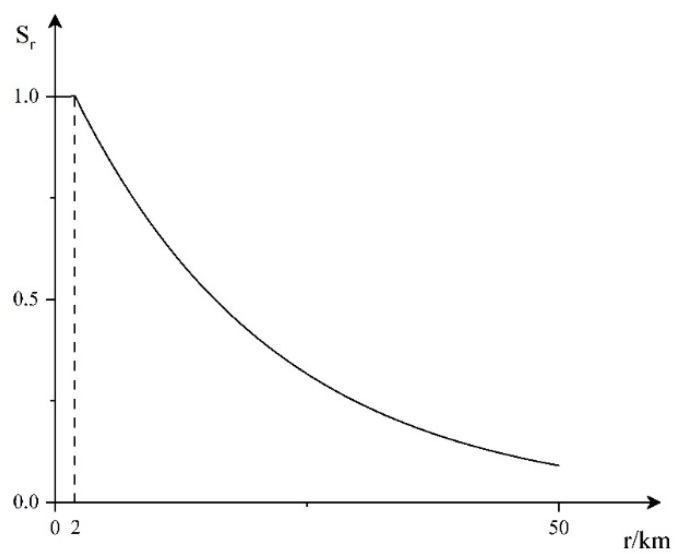
Distance threat function.

**Figure 5 entropy-23-01292-f005:**
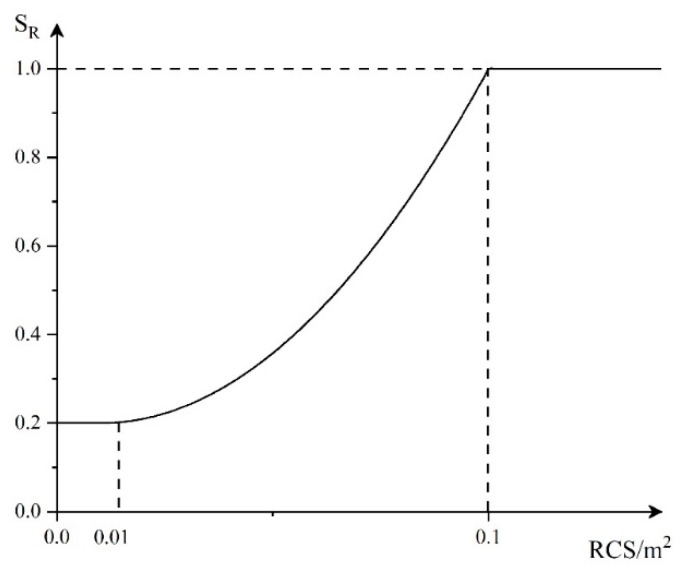
RCS threat function.

**Figure 6 entropy-23-01292-f006:**
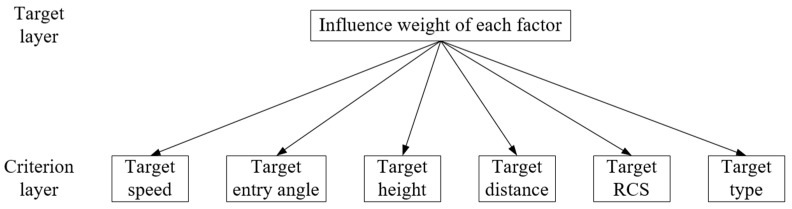
Analytic hierarchy process model diagram.

**Figure 7 entropy-23-01292-f007:**
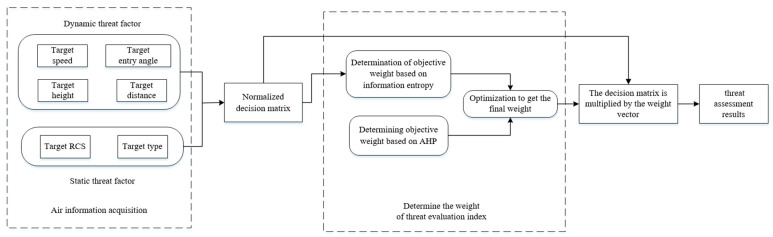
Threat assessment model.

**Figure 8 entropy-23-01292-f008:**
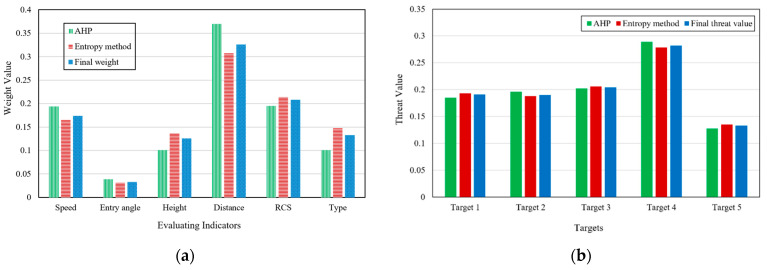
Threat assessment results when partiality coefficient α=0.3. (**a**) Weight value of each evaluation indicator; (**b**) Threat value of targets.

**Figure 9 entropy-23-01292-f009:**
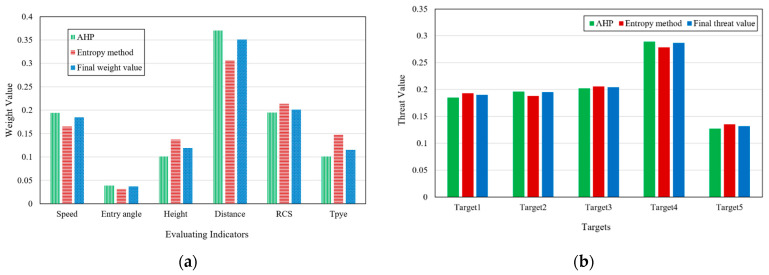
Threat assessment results when partiality coefficient α=0.7. (**a**) Weight value of each evaluation indicator; (**b**) Threat value of targets.

**Table 1 entropy-23-01292-t001:** Analysis of LSS target detection capability.

Detection Means	Radar Detection	Infrared Acquisition
Detection Capability	The detection speed is fast, and it can detect all-weather and long-distance. The effective detection distance is about 5 km. According to the target size and structural characteristics, the reliable detection distance is 1~8 km.	It can detect all weather, but it is difficult to identify weak and small targets, and the detection distance is affected by atmospheric visibility. Under the condition of visibility of 15 km, the maximum detection distance can reach 3.6 km.

**Table 2 entropy-23-01292-t002:** Target attribute comparison matrix.

	Speed	Entry Angle	Height	Distance	RCS	Type
Speed	1	5	2	1/2	1	2
Entry Angle	1/5	1	1/3	1/7	1/5	1/3
Height	1/2	3	1	1/4	1/2	1
Distance	2	7	4	1	2	4
RCS	1	5	2	1/2	1	2
Type	1/2	3	1	1/4	1/2	1

**Table 3 entropy-23-01292-t003:** Target feature information.

Target	Evaluation Indicators					
	Speed/Ma	Entry Angle/(°)	Height/(m)	Distance/ (km)	RCS/(m^2^)	Type
Target 1	0.2	−20	200	2	0.02	rotor
Target 2	0.4	15	500	20	0.05	fixed-wing
Target 3	0.5	45	400	35	0.15	fixed-wing
Target 4	0.3	−55	800	40	0.05	rotor
Target 5	0.2	60	350	5	0.04	rotor

## Data Availability

Not applicable.

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
