# Peer review of "Threat Assessment Method of Low Altitude Slow Small (LSS) Targets Based on Information Entropy and AHP"

_entropy, 2021, doi:10.3390/e23101292_

Round 1

Reviewer 1 Report

The article considers the multi-attribute decision making problem of Low Altitude Slow Small (LSS) target threat. The authors utilize AHP, information entropy, and optimization to determine optimal weights of the assessment criteria. The article is a more or less straightforward application of the combination of existing methods, and for this reason it would be essential that the practical aspects needs to be presented with as much details as possible. Most of my detailed comments relate to this as follows:

  • Could you describe, if possible, and not confidential, what extensive data was used, as mentioned in the beginning of section 2, for determining detection capabilities in Table 1 
  • In my opinion that most important thing that is missing is the justification for parameter choices in the functions quantifying the different indicators as presented in Section 2. While the authors justify some generic directions (e.g. the samller the angle, the greater the threat), but they do not explain the choice of function, e.g. why quadratic, exponential, etc., and how the parameters of the different functions are determined. Are they based on recommendations from the literature, or based on some analysis the authors done on some available data, or the opinion of some experts? These functions are the core of the proposed system, so it would be essential to know exactly how they were chosen.
  • Similarly, more information could be provided about the presented comparison matrix in Section 3. How many expert were involved in creating the matrix, what background they have? Was there any iteration in creating the matrix, did the experts disagree, what values are the most uncertain in the matrix etc? The authors could provide as many details as it is available and possible to present in the article.
  • Is the data presented in Section 5 comes from expert, i.e. is it a realistic example? The presented calculations seem to be correct, but a detailed sensitivity analysis would be very useful. Ideally involving also the parameters in the threat factor quantifying functions, but at least with respect to the bias parameter. Why was 0.5 selected? How do the results change when this value is increased/decreased? 
  • Furthermore, from Figure 9, what we can see that the order of targets is the same independent of the evaluation methods used, and for any target the three values are very close to each other. Why would we then make use of this complex method, instead of simply using either subjective or objective approach, as we do not seem to gain extra insights by combining them?
  • Please define abbreviations at first use (for example UAV is never defined, I suppose it Unmanned Aerial Vehicle)

As mentioned above, since the paper presents an applications without any novel developments on methodologies, and currently the authors do not present much details and explanations on the specific details of the application; this should be addressed before the paper can be considered for publication.

Author Response

Dear reviewer,

Thank you very much for your efforts for my manuscript.

We have revised the manuscript according to your comments.Please see the attachment.

Kind regards,

Mr. Ruining Luo

Reviewer 2 Report

Major remarks:

1) The article lacks a discussion on the comparison of objective and subjective weights. When is it better to use subjective weights and when is it better to use subjective weights? Which are better in certain situations? Can the AHP method be used without any remarks? In this context, I would like to remind you about the disadvantages of AHP related to: the rank reversal problem, the comparison scale [1], the priorities derivation method [2].

2) A literature review should be carried out on other prioritization / weighting methods, e.g. BWM, ANP, REMBRANDT, MACBETH, etc.

Minor remarks:

1) The aim of the research should be clearly stated in the Introduction. Moreover, a structure of the article should be discussed in the last paragraph of the Introduction.

2) The Conclusion section should be improved. Please clear highlight your theoretical and practical scientific contribution. Additionally, conclusions section should be supplemented with potential applications of the results of the conducted research as well as future research directions.

3) If the AHP method is used, at least one publication of its creator - Saaty - should be referred to, e.g. [3]

4) There are very few references in the article. They should be supplemented, incl. indicating that similar topics are covered in Entropy journal.

[1] Ishizaka, A., Lusti, How to derive priorities in AHP: a comparative study. Central European Journal of Operations Research 2006, 14(4), 387-400. https://doi.org/10.1007/s10100-006-0012-9

[2] Ziemba P., Wątróbski J., Jankowski J., Piwowarski M. Research on the Properties of the AHP in the Environment of Inaccurate Expert Evaluations. In: Nermend K., Łatuszyńska M. (eds) Selected Issues in Experimental Economics. Springer Proceedings in Business and Economics. Springer, Cham 2016. https://doi.org/10.1007/978-3-319-28419-4_15

[3] Saaty T.L. The Analytic Hierarchy Process. McGraw-Hill, New York 1980.

Author Response

(The authors gave the same response as above.)

Reviewer 3 Report

The article as it stands looks like a simple application of well-known methods. The description of the main contention and motivation for the study should be improved. Why such methods were chosen. What is the novelty of the proposed approach? There are fresh approaches in the literature such as: 'Best-Worst method and Hamacher aggregation operations for intuitionistic 2-tuple linguistic sets' or 'On the Analytic Hierarchy Process Structure in Group Decision-Making Using Incomplete Fuzzy Information with Applications'. In which the proposed method is better. What are the practical implications? what are the limitations? The literature needs to be extended. Further questions may arise in the next stage of the review. 

Author Response

(The authors gave the same response as above.)

Round 2

Reviewer 1 Report

The authors have considered all the review comments and improved the paper, I recommend it to be accepted for publication.

Author Response

Dear reviewer,

I have received your second review comments. Thank you for your support and recognition. Your comments have played a vital role in the improvement of the manuscript. Thank you again for your efforts.

Kind regards,

Mr. Ruining Luo

Reviewer 2 Report

The paper has been improved and can be published.

Author Response

(The authors gave the same response as above.)

Reviewer 3 Report

As in the previous review, I suggest more emphasise originality and novelty of the used approach.  The authors should in more detail explain:

-What are the practical implications?

-What are the limitations?

-The literature has been extended, however, still needs to be extended.

Further questions may arise in the next stage of the review. 

Author Response

Dear reviewer,

I have received your second review comments. Thank you for your serious and responsible guidance and help.Your valuable comments have played a very important role in improving the manuscript. We have carefully revised it according to your requirements. If there are still deficiencies, please point out and we will continue to improve according to your requirements.

Thank you again for your serious responsibility and efforts.

Kind regards,

Mr. Ruining Luo

Round 3

Reviewer 3 Report

The abstract must absolutely be corrected. Starting with language errors like the first sentence instead of 'has' should be 'have'. The abstract must show the motivation for the research and then indicate what the contribution is. Again, the abstract does not indicate what is the main novelty of the article.

Author Response

Dear reviewer,

Thank you again for your serious responsibility and efforts to my manuscript.

I have received and carefully read your third comment. You highlighted the research motivation, contribution and novelty that should be reflected in the abstract. Thank you very much for your wise comments. We have completely revised the abstract based on your comments. Please refer to the newly submitted manuscript for modifications.

Kind regards,

Mr. Ruining Luo